# The Characteristics and Expression Analysis of the Tomato SlRBOH Gene Family under Exogenous Phytohormone Treatments and Abiotic Stresses

**DOI:** 10.3390/ijms25115780

**Published:** 2024-05-26

**Authors:** Yuanhui Wang, Zesheng Liu, Long Li, Xuejuan Pan, Kangding Yao, Wenying Wei, Weibiao Liao, Chunlei Wang

**Affiliations:** College of Horticulture, Gansu Agricultural University, Yinmen Village, Anning District, Lanzhou 730070, China; 18719851354@163.com (Y.W.); lzs0724@163.com (Z.L.); 15877595464@163.com (L.L.); panxj@st.gsau.edu.cn (X.P.); 19119882925@163.com (K.Y.); 15097284649@163.com (W.W.); liaowb@gsau.edu.cn (W.L.)

**Keywords:** respiratory burst oxidase, abiotic stress, tissue expression profiling, gene family

## Abstract

Respiratory burst oxidase homologs (RBOHs), also known as NADPH oxidases, contribute significantly to the production of ROS in plants, alongside other major sources such as photosynthesis and electron transport in chloroplasts. It has been shown that plant RBOHs play an active role in plant adversity response and electron transport. However, the phylogenetic analysis and characterization of the SlRBOH gene family in tomatoes have not been systematically studied. This study identified 11 SlRBOH genes in the tomato genome using a genome-wide search approach. The physicochemical properties, chromosomal localization, subcellular localization, secondary structure, conserved motifs, gene structure, phylogenetics, collinear relationships, cis-acting elements, evolutionary selection pressures, tissue expressions, and expression patterns under exogenous phytohormones (ABA and MeJA) and different abiotic stresses were also analyzed. We found that the SlRBOHs are distributed across seven chromosomes, collinearity reflecting their evolutionary relationships with corresponding genes in *Arabidopsis thaliana* and rice. Additionally, all the SlRBOH members have five conserved domains and 10 conserved motifs and have similar gene structures. In addition, the results of an evolutionary selection pressure analysis showed that SlRBOH family members evolved mainly by purifying selection, making them more structurally stable. Cis-acting element analyses showed that SlRBOHs were responsive to light, hormone, and abiotic stresses. Tissue expression analysis showed that *SlRBOH* family members were expressed in all tissues of tomato to varying degrees, and most of the *SlRBOHs* with the strongest expression were found in the roots. In addition, the expressions of tomato *SlRBOH* genes were changed by ABA, MeJA, dark period extension, NaCl, PEG, UV, cold, heat, and H_2_O_2_ treatments. Specifically, *SlRBOH4* was highly expressed under NaCl, PEG, heat, and UV treatments, while *SlRBOH2* was highly expressed under cold stress. These results provide a basis for further studies on the function of SlRBOHs in tomato.

## 1. Introduction

Plants are continually exposed to biotic and abiotic stresses, which negatively affect their growth and yield, causing enormous losses in agriculture worldwide. These stressors, such as drought, extreme temperatures, and salt, lead to the over-accumulation of reactive oxygen species (ROS) [1]. The excessive accumulation of ROS causes membrane damage, protein oxidation, and DNA lesions, and can even lead to irreparable metabolic dysfunctions and cell death [2]. ROS are signaling molecules that include singlet oxygen (^1^O_2_), superoxide anion (O_2_^−^), hydrogen peroxide (H_2_O_2_), and a hydroxyl radical (HO^−^) [3]. The controlled generation of ROS is pivotal for normal plant development and adaptation to changes in the external milieu. One of the significant synthesis enzymes of ROS in plants is plasma-membrane localized NADPH oxidases [4]. NADPH oxidase was first discovered in mammals, located on the plasma membrane of phagocytic cells [5]. It is a multi-enzyme complex consisting of six subunits, gp91^phox^, p22^phox^, p47^phox^, p67^phox^, p40^phox^, and Rac, and carries cytochrome C and FAD group composition. Plant NADPH oxidase is a homolog of the mammalian NADPH oxidase catalytic subunit gp91^phox^, which is known as the respiratory burst oxidase homolog (RBOH) [6]. In plants, RBOHs catalyze the conversion of dioxygen (O_2_) to the superoxide anion (O_2_^–^) [4], which suggests that *SlRBOH* may be involved in stress-induced ROS generation when tomato is exposed to various stressors.

Plant RBOHs belong to a multigene family [7]. The first *RBOH* gene in plants was identified in rice, followed by successive identifications in many plants, including *Arabidopsis thaliana* [8], eggplant [3], potato [9], barley [4], maize [10], tobacco [11], and grape [12]. The typical plant RBOH possesses four conserved domains, namely NADPH_Ox, Ferric_reduct, FAD_binding_8, and NAD_binding_6 domain [13]. Members of the RBOH family have different physiological functions in different plant tissues [3]. For example, *AtRBOHC* is involved in root hair growth in *A. thaliana*, whereas *AtRBOHD* and *AtRBOHF* lead to the localized accumulation of superoxide in *A. thaliana* roots, which inhibits lateral root development [14]. *OsRBOHA* is highly expressed in seeds and pollen. Simultaneously, *OsRBOHA* participates in the germination process of rice seeds, and the mutation of OsRBOHA results in reduced pollen viability and seed fertility [15]. In addition, plant RBOHs take part in many kinds of abiotic stress responses. Of the seven *CrRBOH* genes in citrus, five *CrRBOH* genes are responsive to cold stress, and the knock-down of *CsRBOHD* reduces the plant’s cold tolerance [16]. CsRBOH is involved in the maintenance of cold domestication in cucumbers [17]. In rice, drought stress induces RBOH oxidase activity. The overexpression of *OsRBOHA* and *OsRBOHB* enhances plant drought tolerance [18]. Additionally, a mutation of *OsRBOHB* decreases the production of endogenous ROS, and ABA concurrently, which causes a larger stomatal aperture and a reduced drought tolerance. Current studies have shown that the function of plant *RBOH* genes mainly focuses on two aspects, including participating in stress responses and regulating growth and development [19,20].

Tomato is an important economic crop across the world, as well as an important experimental model plant. RBOH-produced ROS plays important roles in plant stress responses and the regulation of plant growth and development. However, familial analyses of the *RBOH* gene in tomato have not been reported. In order to better understand the role of the *RBOH* gene family in tomato, we have studied the physicochemical properties, subcellular localization prediction, secondary structure analysis, conserved motifs, chromosomal localization, *cis*-acting elements, phylogeny, selection for evolutionary pressures, and collinearity analysis of SlRBOH members. We also analyzed the expressions of tomato *SlRBOH* genes in various tissues, exogenous phytohormones, and abiotic stress treatments. This study identifies *SlRBOHs* that play essential roles in different tissues and stresses based on their expression. It provides a basis for subsequent research and the utilization of *SlRBOH* genes.

## 2. Results

### 2.1. Identification of SlRBOH Gene Family Members in Tomato and Analysis of Chromosomal Localization

Ten protein sequences (e = 1 × 10^−5^) containing RBOH domains were retrieved from the *A. thaliana* genome database. Tomato genome annotation files (GFF and FASTA files) were obtained from NCBI (https://www.NCBI.nlm.nih.gov/ (accessed on 5 October 2023)). The tomato genome annotation information and TBtools software were used to compare the RBOH protein sequences in *A. thaliana*, and 11 members of the tomato *SlRBOH* gene family were identified (Figure 1A). Using tomato genome annotation information and TBtools software, the chromosome distribution of *SlRBOH* gene family members was visually analyzed. The tomato *SlRBOH* genes were unevenly distributed on seven chromosomes and were named *SlRBOH1~SlRBOH11* in order (Figure 1B). The number of genes on each chromosome is independent of chromosome size. Based on the evolutionary relationship between different species, we also explored the collinearity relationship between the tomato *SlRBOH* gene and the related genes in *A. thaliana* and rice (Figure 1C). The results of a collinearity analysis showed that 11 genes of tomato were collinear with 10 *A. thaliana* genes and 9 rice genes. There are nine homologous pairs between tomato and *A. thaliana* and five homologous pairs between tomato and rice. These results indicate that tomato *SlRBOH* is closely related to *RBOH* members in *A. thaliana* and rice.

### 2.2. Analysis of Physicochemical Properties and Subcellular Localization of the Tomato SlRBOH Family Members

The Expasy online website (https://web.expasy.org/compute_pi/ (accessed on 8 October 2023)) was used to analyze the physicochemical properties of SlRBOHs (Table 1). The number of amino acids in tomato SlRBOH family members ranges from 776 in SlRBOH11 to 989 in SlRBOH10. The molecular weight of these proteins varies between 89.06 kDa and 112.28 kDa. All tomato SlRBOH proteins have an isoelectric point (pI) greater than 7, indicating that they are iodine proteins. The instability index of these proteins ranges from 37.49 in SlRBOH2 to 50.71 in SlRBOH4, and the aliphatic index varies from 80.37 in SlRBOH6 to 91.04 in SlRBOH4. The hydrophilicity values indicate that all tomato SlRBOH proteins are hydrophilic, with SlRBOH2 having the most negative value at −0.154. Signal peptide prediction results show that no SlRBOH family members contain signal peptide sites. Transmembrane structure prediction results show that all members of the SlRBOH family proteins have a transmembrane domain.

The WoLF PSORT online website (https://wolfpsort.hgc.jp/ (accessed on 8 October 2023)) was used to analyze the subcellular localization of SlRBOH (Table 2). Subcellular localization prediction results showed that the SlRBOH members were mainly distributed in the plasma membrane, endoplasmic reticulum, nucleus, peroxisome, chloroplast, mitochondrion, and cytosol. However, the distribution of each member is different in different cell sites. SlRBOH family members were mostly located in the plasma membrane of the cells, six members were located on the endoplasmic reticulum, five members were located on the nucleus, five members were located on the peroxisomes, four members were located on chloroplasts, four members were located on mitochondrion, and one member was located on the cytosol.

### 2.3. Structural Analysis and Phylogenetic Tree Analysis of the SlRBOH Gene Family

We used TBtools software to visualize the gene structure of the *SlRBOH* gene (Figure 2). The results showed that the number of introns and exons of tomato *SlRBOH* family members were similar in number and distribution, with the number of introns ranging from 11 to 14 and the number of exons ranging from 12 to 15. Notably, *SlRBOH3* and *SlRBOH4* have similar and longer introns, *SlRBOH7* and *SlRBOH8* have similar and longer introns, and *SlRBOH9* and *SlRBOH10* have similar and longer introns, while the other members have shorter introns.

In order to further understand the phylogenetic process of tomato SlRBOH family members, we compared the full-length sequences of 38 RBOH proteins in tomato, *A. thaliana*, rice, and eggplant, and jointly constructed a phylogenetic tree (Figure 3). According to the homology, RBOH proteins in these plants were divided into six subgroups. Group A had nine members, including SlRBOH2 and SlRBOH5. Group B had nine members, including SlRBOH3, SlRBOH4, SlRBOH7, and SlRBOH8. Group C consisted of seven members, including SlRBOH9 and SlRBOH10. Group D consisted of four members, including SlRBOH6 and SlRBOH11. There were four members in Group E, none of which were tomato SlRBOH members. Group F has five members, including SlRBOH1. SlRBOH3 and SlRBOH4, SlRBOH7 and SlRBOH8, and SlRBOH9 and SlRBOH10 had the highest homology coefficients. SlRBOHs and SmRBOHs are the closest relatives.

### 2.4. Conserved Motifs of the Tomato SlRBOH Family

We identified 10 conserved motifs in tomato SlRBOHs. All SlRBOH members have 10 conserved motifs (Figure 4A). The amino acid sequences of the different conserved motifs are represented by a series of letters at each location (Figure 4B). The length of Motif 10 is 41, and the length of the remaining 9 motifs is 50 (Table 3).

### 2.5. Analysis of the Protein Secondary Structure of Tomato SlRBOH Family

In tomato SlRBOH proteins, there exist four kinds of protein secondary structure, including alpha helix (43.23–50.90%), extended chain (12.93–16.76%), a beta of turns (2.71–5.42%), and random coil (32.35–38.90%) (Table 4). The most abundant protein secondary structures among tomato SlRBOH members are mainly α-helices and random coils.

### 2.6. Analysis of Cis-Acting Elements of Tomato SlRBOH Family Genes

We analyzed the promoter sequences (from −2000 bp to −1 bp) of 11 *SlRBOH* genes to detect the *cis*-acting elements (Figure 5). The tomato *SlRBOH* genes contain 22 homologous elements (Table 5). Further investigating the *cis*-elements in the *SlRBOH* promoter sequence, we identified three significant *cis*-elements, including light, stress, and hormone action elements (Figure 6). Twelve elements (MRE, I-box, AE-box, G-box, G-Box, Box 4, GA-motif, GT1-motif, TCT-motif, AT1-motif, GATA-motif, and TCCC-motif) were associated with light response. Four elements (LTR, MBS, ARE, and TGA-element) were associated with stress response, and six elements (P-box, CGTCA-motif, ABRE, TGACG-motif, TCA-element, and TATC-box) were associated with stress response. Except for *SlRBOH1*, the Box4 element was distributed in all the other tomato *SlRBOH* genes and was the most abundant in *SlRBOH2*. The TCCC-motif is the least photoresponsive element. Except for *SlRBOH2*, ARE elements are distributed in the tomato *RBOH* gene, TGA-element is the element with the lowest stress response, and TGA-element exists only in *SlRBOH7*, *SlRBOH8*, and *SlRBOH11*. Except for *SlRBOH1* and *SlRBOH2*, CGTCA-motif and TGACG-motif elements are distributed in the remaining members and in the same number. Therefore, the relative abundance of *cis*-elements associated with light, stress, and hormones suggests that the tomato *SlRBOH* genes may play a key role in regulating plant growth and hormone response.

### 2.7. Evolutionary Selection Pressure Analysis of Tomato SlRBOH Genes

Ka denotes the nonsynonymous substitution rate, that is, the incidence of nonsynonymous mutations in the gene sequence [21]. Ks denotes the synonymous substitution rate, that is, the incidence of synonymous mutations in a gene sequence. The Ka/Ks values can be used to measure the relative frequency of nonsynonymous and synonymous mutations, thus reflecting the selective pressure on genes during evolution [22]. It plays an important role in the evolutionary analysis of gene families. We conducted a Ka/Ks analysis of the tomato *SlRBOH* gene family and found that the Ka/Ks values of *SlRBOH* family genes were all less than 1 (Figure 7). This suggests that the selection pressure of the *SlRBOH* gene family evolved in tomato species is mainly due to purification selection.

### 2.8. Tissue-Specific Expression Analysis of Tomato SlRBOH Genes

To determine the expression specificity of tomato *SlRBOH* genes in different growth stages and tissues, qRT-PCR was used to detect the expression levels of tomato *SlRBOH* genes in different tissues during the nutritive and reproductive growth stages. During the nutritive growth stage, all members were highly expressed in roots except *SlRBOH7* (Figure 8). *SlRBOH1*, *SlRBOH3*, *SlRBOH4*, *SlRBOH8*, and *SlRBOH10* had the highest expression in roots. *SlRBOH2*, *SlRBOH5*, *SlRBOH7*, and *SlRBOH9* had the highest expression in stems. *SlRBOH6* and *SlRBOH11* had the highest expression in leaves. During the reproductive growth period, all *SlRBOHs* were highly expressed in roots and red fruits (Figure 9). Among them, *SlRBOH5* was highly expressed in stems. *SlRBOH2*, *SlRBOH9*, and *SlRBOH10* were highly expressed in leaves. *SlRBOH1*, *SlRBOH3*, *SlRBOH6*, *SlRBOH7*, *SlRBOH8*, and *SlRBOH11* were highly expressed in flowers. *SlRBOH4* was highly expressed in green fruits. The results suggest that *SlRBOH* genes may play unique roles in different growth stages and different tissues of plants.

### 2.9. Expression Analysis of Tomato SlRBOH Genes under Hormonal and Abiotic Stress

To investigate the expression level of *SlRBOH* genes under the effect of hormones, we studied the expression levels of 11 *SlRBOH* genes under 100 μM ABA (Figure 10A) and 50 μM MeJA (Figure 10B) treatments. In the members of the *SlRBOH* family, under ABA treatment for 6 h, the expression levels of all members were significantly up-regulated except *SlRBOH2* and *SlRBOH7*, with *SlRBOH3* having the highest expression level. All members were significantly up-regulated under ABA treatment for 12 h, with *SlRBOH6* having the highest expression level. Under ABA treatment for 24 h, the expression levels of *SlRBOH2*, *SlRBOH3*, *SlRBOH4*, *SlRBOH6*, and *SlRBOH7* were significantly up-regulated. In contrast, the expression levels of the remaining members were significantly suppressed, with *SlRBOH1* having the lowest expression level. Under the effect of MeJA, the expression levels of *SlRBOH1*, *SlRBOH4*, *SlRBOH6*, and *SlRBOH7* were significantly increased, while the expression levels of other members were significantly inhibited.

We investigated the expression level of *SlRBOH* genes under the effect of different abiotic stresses. We studied the expression levels of 11 *SlRBOH* genes under 200 mM NaCl, PEG, cold (4 °C), heat (40 °C), H_2_O_2_, 253.7 nm UV, and dark period extension (Figure 11A–G) treatments. In the members of the *SlRBOH* family, under NaCl treatment, the expression levels of *SlRBOH3*, *SlRBOH4*, *SlRBOH6*, and *SlRBOH7* were significantly up-regulated over 6–24 h, while the expression levels of *SlRBOH5* were significantly inhibited over 6–24 h. Under PEG treatment for 6 h, the expression level of *SlRBOH11* was significantly inhibited. Under PEG treatment for 12 h, the expression level of all members was significantly up-regulated, and the expression level of *SlRBOH4* was the highest. Under PEG treatment for 24 h, in addition to *SlRBOH2* and *SlRBOH5*, the expression level of other members was up-regulated, and the expression level of *SlRBOH4* was the highest. *SlRBOH5* expression levels were inhibited from 6 to 24 h. Under cold (4 °C) treatment for 6 h, the expression levels of *SlRBOH1*, *SlRBOH2*, and *SlRBOH4* were significantly up-regulated. In contrast, the expression levels of other members were significantly decreased, among which the expression levels of *SlRBOH6* and *SlRBOH11* were most significantly down-regulated. Under cold (4 °C) treatment for 12–24 h, the expression levels of *SlRBOH2* and *SlRBOH6* were significantly up-regulated, while the expression levels of other members were significantly inhibited. Cold (4 °C) treatment reduced the transcription levels of *SlRBOH5*, *SlRBOH8*, and *SlRBOH11*. Under heat (40 °C) treatment, the expression levels of *SlRBOH9* and *SlRBOH11* were significantly inhibited at 6–24 h, while the expression levels of other members were significantly increased at 6 h. The expression level of *SlRBOH4* was the highest at 6–24 h. Under H_2_O_2_ treatment, the expression levels of all members were significantly up-regulated at 6 h. Except for *SlRBOH9* and *SlRBOH10*, the expression levels of all members were significantly up-regulated at 6–24 h. Under UV treatment for 6 h, the expression levels of *SlRBOH1*, *SlRBOH2*, *SlRBOH3*, and *SlRBOH4* were significantly up-regulated, while the expression levels of other members were significantly inhibited. Under UV treatment for 12 h, except for *SlRBOH6*, *SlRBOH9*, and *SlRBOH10*, the expression level of other members was significantly up-regulated, and the expression level of *SlRBOH1* was the highest. The expression levels of all members were significantly up-regulated at 24 h. Under dark period extension treatment for 6–24 h, the expression levels of *SlRBOH4*, *SlRBOH5*, and *SlRBOH6* were significantly up-regulated, while the expression levels of other members were significantly decreased. Under dark period extension treatment for 24 h, the expression levels of all members were significantly inhibited at 24 h.

## 3. Discussion

ROS play a key role in signal transduction in cells [23]. They are involved in regulating growth, development, responses to environmental stimuli, and cell death. RBOHs are key signaling nodes in the ROS network [23]. Plant RBOH gene families encode NADPH oxidases and play important roles in the production of ROS, plant signaling, growth, and stress responses [24]. Benefiting from the availability of whole-genome sequences in recent years, a few model and crop plant NADPH oxidase families have been identified at the genome-wide level [25]. For example, previous studies identified 10 *AtRBOHs* in *A. thaliana* [26], 9 *OsRBOHs* in rice [27], and 8 *SmRBOHs* members in eggplant [3]. In this study, the tomato *SlRBOH* family genes were bioinformatically analyzed and 11 *SlRBOH* genes were identified (Table 1), more than that of *A. thaliana*. As the differences in genome size between species may lead to variations in the number of family genes, the increase in *SlRBOH* family members may also be due to a larger genome size in tomato than in *A. thaliana* [28]. In addition, genome-wide replication increases the number of genes, which in turn leads to an increase in the number of gene family members [28,29]. In our study, *SlRBOH3* and *SlRBOH4*, *SlRBOH7* and *SlRBOH8*, and *SlRBOH9* and *SlRBOH10* are located on the same chromosome, respectively (Figure 1B). They also have similar gene structures and the same number of conserved motifs (Figure 2 and Figure 4). Phylogenetic analyses revealed the highest homology coefficients between *SlRBOH3* and *SlRBOH4*, *SlRBOH7* and *SlRBOH8*, and *SlRBOH9* and *SlRBOH10* (Figure 3). Combined with a collinearity analysis of RBOH genes in tomato, *A. thaliana*, and rice (Figure 1C), we found that there were homologous gene pairs between *SlRBOH2*, *SlRBOH6*, and *SlRBOH11* and members of both the *A. thaliana* and rice *RBOH* families. Meanwhile, there were no homologous gene pairs between *SlRBOH3*, *SlRBOH4*, *SlRBOH8*, and *SlRBOH9*, nor in *A. thaliana* or rice. In summary, we speculate that the *SlRBOH2*, *SlRBOH6*, and *SlRBOH11* genes may have been inherited from earlier plants. In contrast, the *SlRBOH3*, *SlRBOH4*, *SlRBOH8*, and *SlRBOH9* genes may have been derived from the tomato genome replication. Therefore, the increase in SlRBOH members may be related to gene duplication, which is similar to previous studies [10]. There are four typical conserved structural domains in plant RBOHs, including NADPH_Ox, Ferric_reduct, FAD_binding_8, and NAD_binding_6 [10]. The NADPH_Ox domain generates ROS, the C-terminal region contains NAD_binding_6, and FAD_binding_8 domains play crucial roles in electron transfer and ROS production [24,30]. Ferric-reductase is related to iron reductase [3]. All RBOHs contain the NADPH_Ox structural domain, whereas the other three typical structural domains may be incomplete in different plant species. In *A. thaliana*, all ten *AtRBOHs* had the above four typical conserved domains [26]. However, in wheat, only 4 out of 36 NADPH oxidases had the NADPH_Ox domain, but lacked one or two of the other conserved domains [24]. In eggplant, five out of eight *SmRBOHs* lacked the FAD-binding domain, which was substituted by the NOX_Duox_like_FAD_NADP domain [3]. In this experiment, we visualized the structural domains of the tomato *SlRBOH* genes and found that all *SlRBOHs* have five structural domains, including NADPH_Ox, Ferri_reduction, FAD_binding_8, NAD_binding_6, and EF_hand_7 (Figure 1A). Therefore, the tomato SlRBOHs may share a conserved function concerning ROS production with other plant RBOHs. It has been found that the protein containing the EF_hand_7 domain may interact with the small GTPases, which are regulators during RBOH-mediated ROS production. Thus, the GTPases may participate in the ROS synthesis process of ROS by tomato SlRBOH proteins, which needs to be verified further.

We also analyzed the physicochemical properties of tomato SlRBOH members (Table 1). The results showed that all tomato SlRBOH proteins were iodine and hydrophilic proteins, and similar results have been reported in eggplant [3]. Meanwhile, all SlRBOH members have transmembrane structures that transport electrons across biological membranes to reduce oxygen to superoxide anion (·O_2_^−^) [5]. Subcellular location is a crucial characteristic of proteins, and the proteins in different subcellular locations have different functions [31]. Some studies have shown that the RBOHs are predominantly localized to the plasma membrane and function as part of the plasma membrane redox system, transferring electrons from intracellular NADPH/NADH to oxygen, producing a large amount of O_2_^−^ [32]. Subsequently, the O_2_^−^ is disproportionated to produce H_2_O_2_ and other ROS. Our study found that all SlRBOH members were predicted to be predominantly localized at the plasma membrane (Table 2). Hence, we surmise that the membrane-bound SlRBOH proteins may be involved in the ROS production process by a similar mechanism, as reported previously [3,10,25].

Analysis of the gene structure revealed that the exon number of *SlRBOH* was between 12 and 15 (Figure 2), which is similar to that of *A. thaliana* and rice [25]. Many biological studies are based on determining the phylogenetic relationships between species [33]. Phylogenetic analyses showed that the RBOH members in *A. thaliana*, rice, tomato, and eggplant could be classified into six groups (Figure 3). The 11 SlRBOH members were unevenly distributed among the five groups. This suggests that the evolutionary relationship between tomato SlRBOH and *A. thaliana*, as well as rice and eggplant RBOH, is strong. Among them, SlRBOHs and SmRBOHs were the closest to each other regarding their evolutionary relationships, which might suggest a functional similarity between SlRBOHs and SmRBOHs. Additionally, we conducted covariance analyses of tomato, *A. thaliana*, and rice *RBOH* families (Figure 1C). There are more than five homologous pairs between *SlRBOHs* and both *AtRBOHs* and *OsRBOHs*. This suggests that the *RBOHs* family is evolutionarily conserved between plant species. In proteins, alpha helices play a role in supporting and stabilizing the molecular structure [34,35]. In this study, the secondary structure of the SlRBOH family is mainly α-helical (Table 4). Thus, the α-helices may lead to greater structural stability within the SlRBOH family members. Genetic selection pressures contribute to understanding evolutionary relationships. Ka denotes the non-synonymous substitution rate, which is the incidence of nonsynonymous mutations in the gene sequence [21]. Ks denotes the synonymous substitution rate, which is the incidence of synonymous mutations in a gene sequence. Ka/Ks values can be used to measure the relative frequency of nonsynonymous and synonymous mutations, thus reflecting the selective pressure on genes during evolution [22]. The Ka/Ks values less than 1 usually represent that the genes mainly evolve from purification selection. We performed a Ka/Ks analysis on the *SlRBOH* family genes and found that the Ka/Ks values in the *SlRBOH* genes were less than 1 (Figure 7). This indicates that the *SlRBOH* family genes in tomato may evolve mainly from purifying selection, which makes them more evolutionarily conserved, structurally stable, and functionally consistent. We also examined the conserved motifs of the tomato SlRBOH members. We found that all SlRBOHs have 10 conserved motifs and that the 10 motifs of all members are arranged in the same order from the N-terminus to the C-terminus (Figure 4). This suggests that the tomato SlRBOH family members are highly conserved among themselves and may share functional similarities. Similar results have been reported in eggplant [3].

The tissue-specific expression of genes is essential for plant growth and development and provides important insights into understanding gene function [36]. Tissue-specific expression patterns of *RBOHs* have been reported in many plants. In this study, we found that *SlRBOH3*, *SlRBOH4*, *SlRBOH7*, and *SlRBOH8* were strongly expressed in tomato root tissues, which is similar to the findings of *A. thaliana AtRBOHE* [27], barley *HvRBOHG* [37], and grape *VvRBOHE* [12]. Meanwhile, *SlRBOH1*, *SlRBOH6*, and *SlRBOH11* were strongly expressed in tomato flower tissues, and there are similar reports on *A. thaliana AtRBOHH*, barley *HvRBOHD*, and rice *OsRBOHD* [8]. Remarkably, SlRBOH3, SlRBOH4, SlRBOH7, and SlRBOH8 were located in the same branch with AtRBOHE in the phylogenetic analysis (Figure 3). SlRBOH1, SlRBOH6, and SlRBOH11 were located in adjacent branches with AtRBOHH and OsRBOHD. It can be observed that members of the specific RBOH branches in different plant species may exhibit similar expression profiles. Furthermore, our results also suggest that the expression pattern of the same *SlRBOH* gene in the same tissue may be different at different growth periods. *SlRBOH2* was highly expressed in stems during the nutrient growth stage, whereas *SlRBOH2* was highly expressed in leaves during the reproductive growth stage (Figure 8 and Figure 9). In addition, all tomato *SlRBOH* genes were highly expressed in red fruit, similar to findings for strawberry *FvRBOHs* [25]. This indicates that the *SlRBOHs* are comprehensively expressed in different tomato tissues.

*Cis*-acting elements regulate the expression of target genes by binding the trans-acting factors [38]. It has been reported that *RBOHs* respond to environmental stimuli through hormonal signaling networks involving abscisic, salicylic, and jasmine acid, as well as ethylene [39]. In the present study, most *SlRBOHs* contained ABA response elements (ABRE) and MeJA (CGTCA-Motif and TGACG-Motif) response elements, which is similar to the findings in eggplant [3]. Quantitative analyses showed that all *SlRBOHs* responded to ABA and MeJA treatments in various degrees (Figure 10). It has been found that OsRBOHB can be involved in influencing ABA signaling in rice [3]. The mutation of *OsRBOHB* not only repressed ROS production, but also decreased ABA content and enhanced the stomatal aperture, leading to reduced drought tolerance [18]. Therefore, we hypothesize that the function of tomato SlRBOH members may be correlated to ABA and MeJA signals, which needs further identification.

Moreover, the expression of *RBOH* family members in various plants can be stimulated by different abiotic stresses, such as drought [40], salt [37], heat, wounding [41], and cold stress [25]. Meanwhile, many abiotic stress response components (such as ARE, MBS, TGA-element, and LTR) were found in the promoter regions of *SlRBOHs* (Figure 6). In addition, we have found that *SlRBOH1*, *SlRBOH3*, *SlRBOH4*, *SlRBOH6*, *SlRBOH7*, and *SlRBOH8* were significantly up-regulated under both NaCl and PEG treatments (Figure 11). Similarly, *AtRBOHD* and *AtRBOHA* were involved in salt stress response [26], and drought stress significantly stimulated the expression of *OsRBOHB* and *OsRBOHH* [42]. Furthermore, plant *RBOHs* have been found to respond to cold stress [43]. Meanwhile, we found that the expression of *SlRBOH2* was significantly up-regulated under cold treatment, and the remaining members were significantly repressed (Figure 11). In *A. thaliana*, *AtRBOHA* expression was up-regulated under cold stress, whereas the transcript abundances of *AtRBOHD*, *AtRBOHE*, *AtRBOHI*, and *AtRBOHH* were significantly down-regulated [44]. Notably, SlRBOH2 and AtRBOHA are located in the same branch in the phylogenetic analysis (Figure 3). This result further suggests that the RBOH members with high homology may contribute similar functions when plants suffer abiotic stresses, which needs to be studied further. In addition, the cell types in the organ framework cannot be ignored [45]. Some *SlRBOH* genes may be expressed only in trichoblasts, whereas others are expressed in cortex II and III. This cell-specific expression may lead to different resistance and growth effects [46]. Therefore, an in-depth understanding of the expression pattern of *SlRBOH* genes in different cell types is crucial to unravelling the mechanisms involved in stress response and growth promotion in plants. Future studies should focus on the effects of cell-specific expression on plant physiological functions to further explore the mechanism of action of SlRBOH members.

## 4. Materials and Methods

### 4.1. Identification of the SlRBOH Family Members in Tomato

The whole-genome data (SL3.0) and annotation files (SL3.0) for tomato were downloaded from NCBI (https://www.NCBI.nlm.nih.gov/ (accessed on 5 October 2023)). Tomato CDS sequences were extracted through the ‘Gtf/Gff3 Sequences Extract’ feature of TBtools software (v1.09876), and the software’s ‘Batch Translate CDS to Protein’ function was then used to convert CDSs to protein sequences. The *A. thaliana* database was used to search for the identified AtRBOH family members, and the ID and protein sequences of the *A. thaliana* AtRBOH family members were saved. We made a comparison of the protein sequences of all AtRBOHs members of the tomato whole genome sequence using TBtools software to identify the possible members of SlRBOH. To determine further whether an identified protein belonged to the *RBOH* gene family, the ‘Batch Web CD-Search Tool’ function of NCBI and the ‘Visualize NCBI CDD Domain Pattern’ function of TBtools were used to analyze the protein domain, and those that did not contain RBOH domains were deleted. Finally, the members of the tomato *RBOH* gene family were obtained, and these genes were named *SlRBOH* genes.

### 4.2. Physicochemical Properties Analysis of the Tomato SlRBOH Gene Family

To analyze the physicochemical properties of the tomato *SlRBOH* gene family, the molecular weights, instability coefficients, isoelectric points, and hydrophilicity of each member of the horse tomato *SlRBOH* genes were analyzed using the Expasy online website (https://web.expasy.org/compute_pi/ (accessed on 8 October 2023)). The WoLF PSORT online website (https://wolfpsort.hgc.jp/ (accessed on 8 October 2023)) was used to analyze the subcellular localization of each *SlRBOH*. The signal peptide prediction of the tomato SlRBOH protein was performed using SignalP (https://services.healthtech.dtu.dk/service.php?SignalP-5.0 (accessed on 8 October 2023). The transmembrane domain prediction of the tomato SlRBOH protein was performed using SignalP (https://services.healthtech.dtu.dk/service.php?SignalP-5.0 (accessed on 8 October 2023)).

### 4.3. Gene location, Ka (Nonsynonymous)/Ks (Synonymous) Analysis, and Gene Structure Analysis

The chromosomal position distribution of tomato *SlRBOH* genes was analyzed using the ‘Gene Location Visualize from GTF/GFF’ function of TBtools software. The ‘Simple Ka/Ks Calculator (NG)’ function of TBtools software was used to calculate the selection and evolutionary pressure values of the tomato *SlRBOH* gene family. The gene structure of each member in the *SlRBOH* families was analyzed using the ‘Visualize Gene Structure (from GTF/GFF3 File)’ function of TBtools software.

### 4.4. Conserved Motif and Protein Conserved Domain Analysis

The shared conserved motifs of the tomato *SlRBOH* gene family were analyzed online using the MEME website (MEME—Submission form (meme-suite.org) (accessed on 10 October 2023)), and the results were visualized using TBtools software; the structural visualization of the tomato *SlRBOH* gene family was undertaken using the TBtools ‘Gene Structure View (Advanced)’. The conserved structural domains of the tomato *SlRBOH* gene family were analyzed using DNAMAN software (v6).

### 4.5. Phylogenetic Tree and Cis-Acting Elements Analysis

The RBOH protein sequences of *A. thaliana*, rice, and eggplant were downloaded from NCBI (https://www.NCBI.nlm.nih.gov/ (accessed on 5 March 2024)). The protein sequences of tomato, *A. thaliana*, rice, and eggplant were combined in the same file, and the evolutionary tree was constructed using MEGA11 software (v11.0.13), in which the neighbor-joining method was used; the number of replicates was set to 1000, and the rest of the options were set to the default values. The website Evolview (Evolview: login (evolgenius.info) (accessed on 12 October 2023)) was then used for the further modification of the evolutionary tree.

The ‘Gtf/Gff3 Sequences Extract’ and ‘Fasta Extract (Recommended)’ functions of TBtools were used to extract the 2000 upstream *SlRBOH* genes of tomato from the tomato gene databases. Here, 2000 bp of data upstream of the tomato *SlRBOH* genes were extracted from the tomato gene databases; these were submitted to the PlantCARE (http://bioinformatics.psb.ugent.be/webtools/plantcare/ (accessed on 26 February 2024)) database for gene homeotic element (promoter) analysis and were visualized using TBtools.

### 4.6. Collinearity Analysis

The relationship between *A. thaliana*, rice, and eggplant were analyzed. The gene annotation files of all species were from NCBI (https://www.NCBI.nlm.nih.gov/(accessed on 5 March 2024)). We downloaded the relative FASTA and GFF3 files representing genetic relationships and compared the gene annotation files of the three species through the ‘One Step MCScanX’ function in TBtools software. Then, the collinearity analysis was performed using the ‘Text Merge for MCScanX’ function. Finally, the result was visualized using the ‘Multiple Synteny Plot’ function in TBtools software.

### 4.7. Plant Materials and Growth Conditions

‘Micro-Tom’ is a tomato variety widely used in scientific research. ‘Micro-Tom’ is characterized by its small size, short growth cycle, small fruits, and a high degree of self-flowering [47]. ‘Micro-Tom’ tomato seeds with full granules and consistent sizes were selected and placed into a 50 mL centrifuge tube and sterilized with 1% NaClO solution for 10 min. The disinfected seeds were placed in a 250 mL conical bottle filled with 100 mL sterile water and cultured in an HYG-C shaker at a rotating speed of 180 r min^−1^, 25 °C, for 3 days. We changed the sterile water once a day. The germinated tomato seeds were planted in porous trays containing nutrient soil and placed in growing chambers. The light intensity in the growth chamber was 250 μmol photon m^−2^ s^−1^, the environment of the growth room was controlled to have a photoperiod of 16/8 h (light/dark) and an air temperature of 26 ± 2 °C/20 ± 2 °C (day/night). The relative humidity was 60%. The plant material was grown for two weeks and then transplanted from the soil into fresh water and incubated for 2 days for seedling retardation, and after two days, it was switched to a 1/2 Hoagland nutrient solution for further incubation. Eventually, seedlings of uniform size at 21 days of age were selected for follow-up treatment.

### 4.8. Stress Treatments and Tissue Expression

For inducing salt and drought stress, the selected seedlings were transferred into a 1/2 Hoagland nutrient solution containing 200 mM NaCl and 20% (*w*/*v*) PEG6000, respectively, for salt and drought treatments. For ABA, MeJA, and H_2_O_2_ stresses, treatments were applied through foliar spraying with solutions of 100 μM ABA, 50 μM MeJA, and 10% (*w*/*v*) (2.94 M) hydrogen peroxide (H_2_O_2_), respectively. The control group grew in a 1/2 Hoagland nutrient solution without adding other reagents. All seedlings were grown under the same conditions in a growing chamber. For hot and cold treatments, the seedlings were moved to other growth chambers, placed in an environment of 4 °C or 40 °C and cultivated with a 1/2 Hoagland nutrient solution, without adding other reagents. Some of the selected seedlings were transferred to a growth chamber with a UV-C irradiation intensity of 253.7 nm, using UV-C lamps (TUV PL-S 40 W/4P, Philips, Poland), and other growth conditions were the same as the control. The dark period extension treatment began at the end of the dark cycle of the tomato seedlings, which were placed in a dark environment with a 1/2 Hoagland nutrient solution for 24 h. Other growth conditions were consistent with CK. After 0, 6, 12, and 24 h, whole seedlings of each treatment (root, stem, and leaf) were harvested separately, then immediately frozen with liquid nitrogen and stored at −80 °C, respectively. Each treatment contained 3 biological replicates, each containing 8 seedlings. All of the above treatment concentrations are from previous pre-experiments, and the corresponding IC_50_ curves can be found in Appendix A. Also, the roots, stems, and leaves of the 21-day-old untreated seedlings were collected for the *SlRBOH* gene expression analysis of the vegetative growth period. The roots, stems, leaves, and flowers of the untreated plants were collected at the flowering stage (55-day-old seedlings), and the corresponding green fruits (25 days after pollination) and mature fruits (45 days after pollination) were also collected (one fruit selected in each plant) to analyze the *SlRBOH* gene expression levels of the reproductive growth period. Each treatment contained 3 biological replicates, each containing 8 plants.

### 4.9. RNA Isolation and qRT-PCR

Total RNA was extracted from the samples using a TRIzol reagent (Invitrogen, Carlsbad, CA, USA) [48,49]. The purity and concentration of RNA were then examined using a Pultton P100_+_ ultra-micro spectrophotometer (Wuzhou Dongfang, Beijing, China). The A260/A280 ratios of the RNA samples between 2.0 and 2.1 were chosen for the subsequent experiments. Then, the FastQuant First Strand cDNA Synthesis Kit (Tianen, Beijing, China) was used to synthesize cDNA. These reactions were executed under the following conditions: 37 °C for 15 min, 85 °C for 5 s, and finally ending at 4 °C. The SYBR Green Premix Pro Taq HS Premix kit was used for qRT-PCR with a Light Cycler 480 Real-Time PCR System (Roche Applied Science, Penzberg, Germany). The qRT-PCR reaction system contained 10 µL 2 × SYBR Green Pro Taq HS Premix, 0.4 µL primer F, 0.4 µL primer R, 2 µL cDNA, and 7.2 µL ddH_2_O. The primers used in qRT-PCR were designed with Primer Premier 5.0 (Premier Biosoft Corporation, San Francisco, CA, USA), and the internal reference was *SlActin* (NC015447.3), which showed a stable expression level among all the samples tested in our previous study [29,50]. The sequences of the above primers are listed in Appendix A. The 2^−∆∆CT^ calculation method was used to quantify the relative expression of each gene, as described in Schnittger, T.D. [51]. The relative expression values of each gene under each treatment at different treated times were calculated by comparing them with those at 0 h. All experiments in our study were repeated three times independently to ensure the reliability and statistical significance of the results.

## 5. Conclusions

In this study, we identified 11 *SlRBOH* genes in tomato and analyzed their physicochemical properties, chromosomal localization, subcellular localization, secondary structure, conserved motifs, gene structure, phylogenetics, collinear relationships, *cis*-acting elements, evolutionary selection pressures, tissue expressions, and expression patterns under exogenous phytohormones and different abiotic stresses. All *SlRBOHs* are highly conserved among members, and SlRBOHs are most closely related to SmRBOHs in homology analyses. Our study suggests a role of *SlRBOH* genes in tomato growth and plant hormone and abiotic stress responses, which provides a theoretical basis for further exploration of the functions of tomato SlRBOH members. 

## Figures and Tables

**Figure 1 ijms-25-05780-f001:**
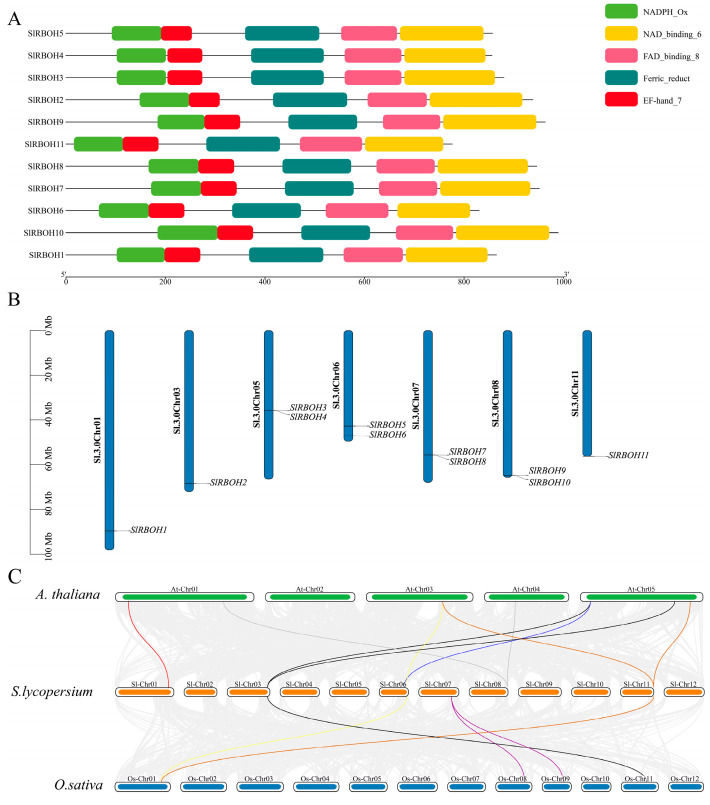
(**A**) Conserved domain of the *SlRBOH* genes in tomato. Domains of the NADPH_Ox, Ferric_reduct, FAD_binding_8, NAD_binding_6 motif, and EF_hand_7. (**B**) Chromosomal localization of the *SlRBOH* gene in tomato. Chromosome positioning was based on the physical location of the 11 tomato *SlRBOHs*. Gene names are indicated in black. The scale bar is on the left. (**C**) Collinearity analysis of *RBOH* gene family in tomato, rice, and *A. thaliana*. The red line indicates the *SlRBOH1* gene, the black line indicates the *SlRBOH2* gene, the blue line indicates the *SlRBOH5* gene, the yellow line indicates the *SlRBOH6* gene, the purple line indicates the *SlRBOH7* gene, the orange line indicates the *SlRBOH10* gene, and the brown line indicates the *SlRBOH11* gene. Gray lines in the background indicate collinear blocks within the tomato, rice, and *A. thaliana* genes.

**Figure 2 ijms-25-05780-f002:**
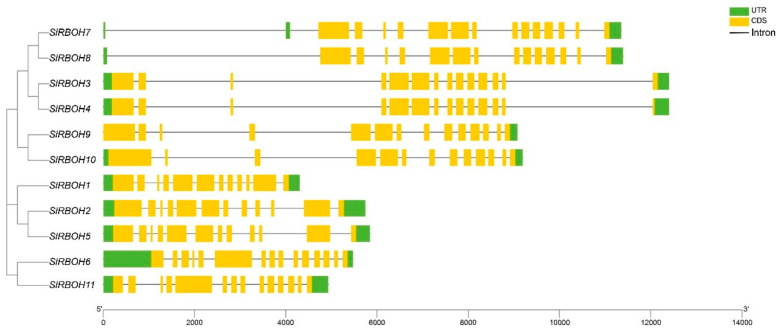
The exon–intron structure of the *SlRBOH* gene family in tomato. The evolutionary tree was constructed based on the full lengths of tomato SlRBOH protein sequences using MEGA11.0 (v11.0.13). The exon–intron graph of tomato *SlRBOH* genes was drawn using TBtools software (v1.09876). The untranslated regions (UTRs) are indicated by thick green boxes. The exons are indicated by thick yellow boxes. The introns are indicated by black lines.

**Figure 3 ijms-25-05780-f003:**
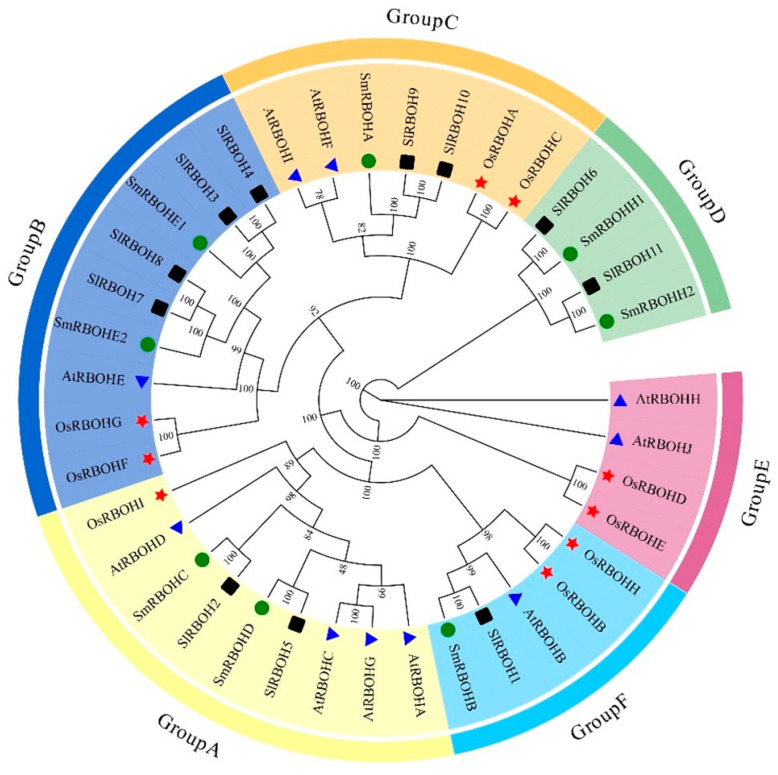
Unrooted phylogenetic tree of tomato, *A. thaliana*, rice, and eggplant *RBOH* gene families. A phylogenetic tree containing 11 tomato, 10 *A. thaliana* (At), 9 rice (Os), and 8 eggplant (Sm) RBOH proteins was constructed using the maximum likelihood method. These six subgroups have different colors. The four different colored shapes represent RBOH proteins from four species. The black square is tomato, the blue triangle is *A. thaliana*, the red star is rice, and the green circle is eggplant. The numbers on the nodes in the phylogenetic tree indicate the percentage of confidence in the bootstrap validation for that branch.

**Figure 4 ijms-25-05780-f004:**
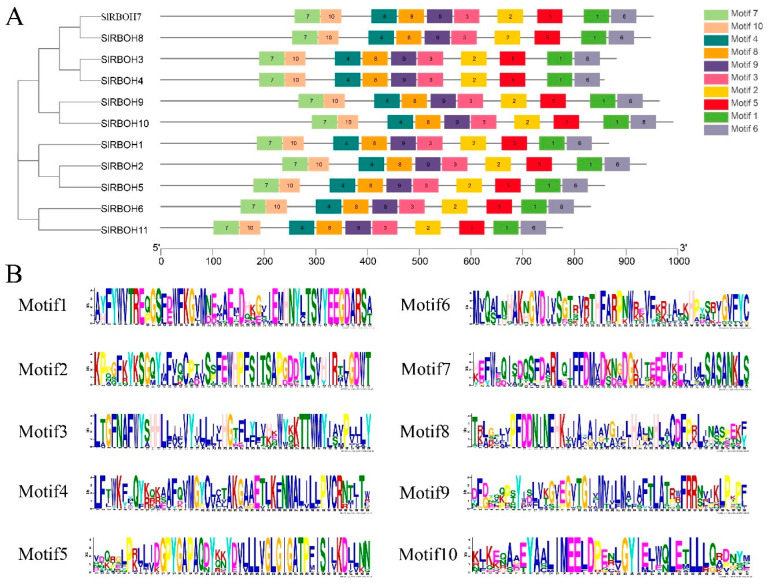
Motif composition and distribution of tomato SlRBOH proteins. (**A**) Colored boxes represent different conserved motifs, and (**B**) 1–10 motifs are shown. (**B**) Amino acid sequences of different conserved motifs, represented by stacked letters at each position. The total height of the stack represents the information content of the relative amino acid in the position of each letter in the motif in bits. The height of the individual letter in a stack was calculated by the probability of the letter at that position times the total information content of the stack. The *X*- and *Y*-axes represent the width and the bits of each letter, respectively.

**Figure 5 ijms-25-05780-f005:**
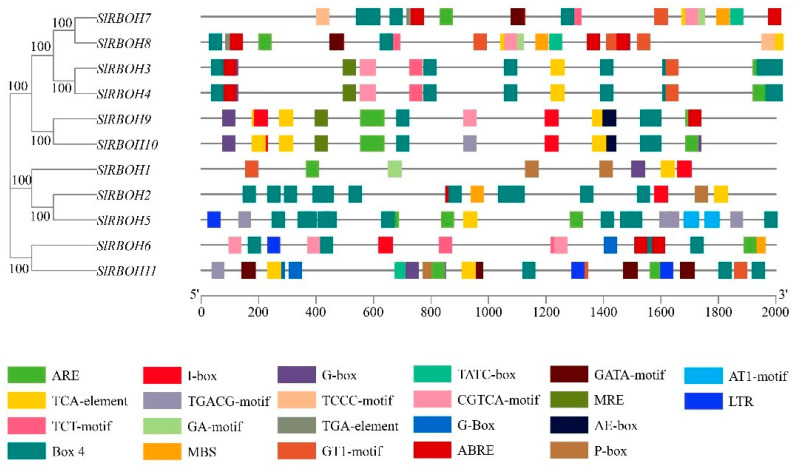
Analysis of *cis*-acting elements of the *SlRBOH* genes family in tomato. Different colored wedges represent different *cis*-elements. The length and position of each *SlRBOH* gene were mapped to scale. The scale bar represents the length of the DNA sequence.

**Figure 6 ijms-25-05780-f006:**
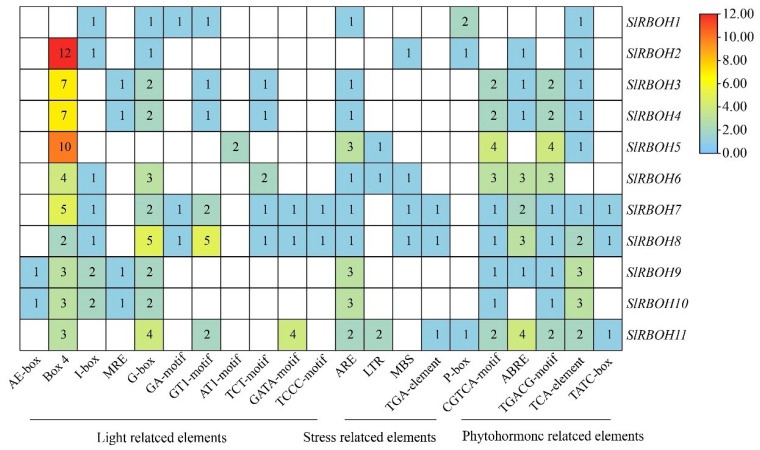
The number of *cis*-acting elements in tomato *SlRBOH* genes. The different colors and numbers of the grid indicate the numbers of different *cis*-acting regulatory elements in these *SlRBOH* genes.

**Figure 7 ijms-25-05780-f007:**
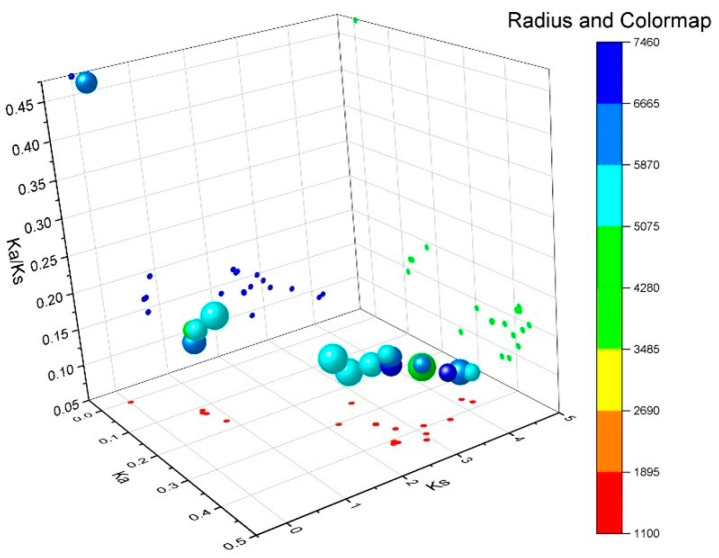
Evolutionary selection pressure analysis of the *SlRBOH* homologous gene pairs. The *X*-axis represents the Ka value, the *Y*-axis represents the Ks value, and the *Z*-axis represents the ratio of Ka to Ks. The color scale represents the fold change normalized by the log2-transformed data.

**Figure 8 ijms-25-05780-f008:**
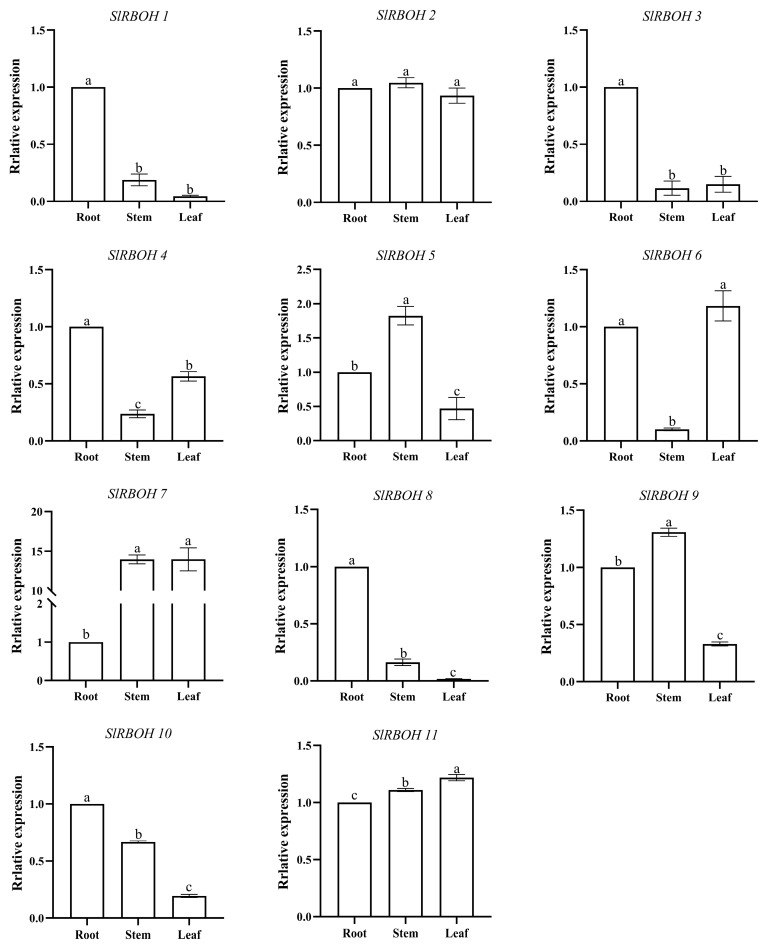
Expression levels of *SlRBOH* genes of different plant tissues in a vegetative growth period. The expression patterns of *SlRBOH1*-*SlRBOH11* in different tissues are shown in A–K, respectively. Error bars represent the standard error (SE) of three replicates. The relative expression of each gene in different tissues is expressed as mean ± SE (*n* = 3). Bars with different lowercase letters were significantly different according to Duncan’s multiple range tests (*p* < 0.05).

**Figure 9 ijms-25-05780-f009:**
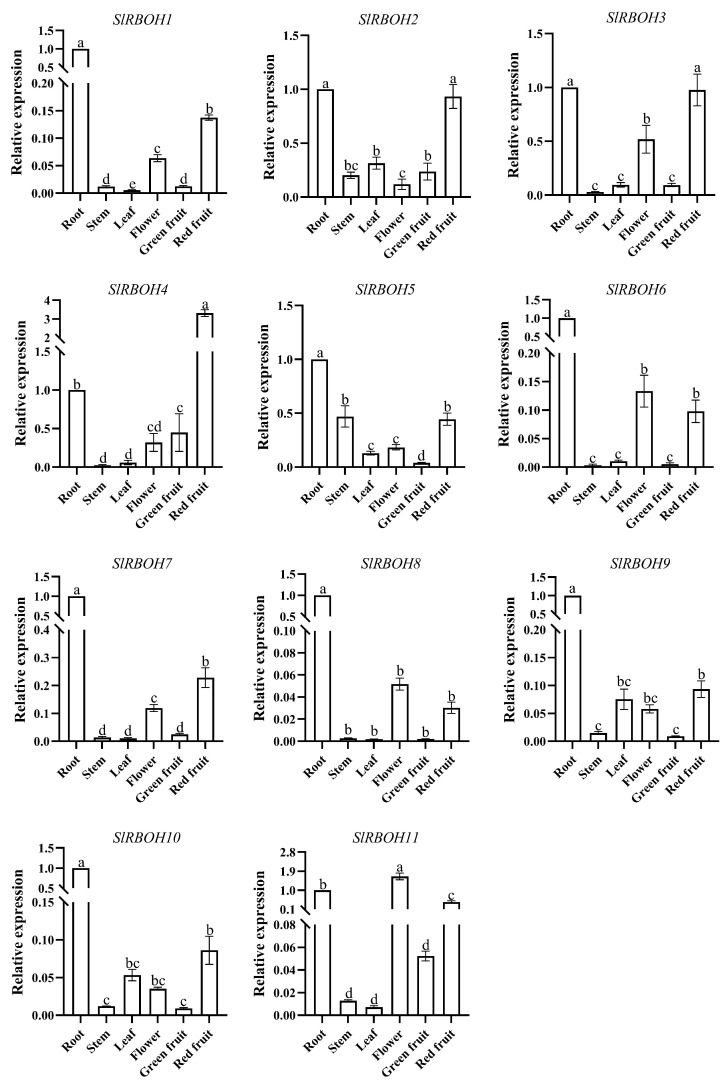
Expression levels of *SlRBOH* genes of different tissues in a reproductive growth period. The expression patterns of *SlRBOH1*-*SlRBOH11* in different tissues are shown in A–K, respectively. Error bars represent the standard error (SE) of three replicates. The relative expression of each gene in different tissues is expressed as mean ± SE (*n* = 3). Bars with different lowercase letters were significantly different according to Duncan’s multiple range tests (*p* < 0.05).

**Figure 10 ijms-25-05780-f010:**
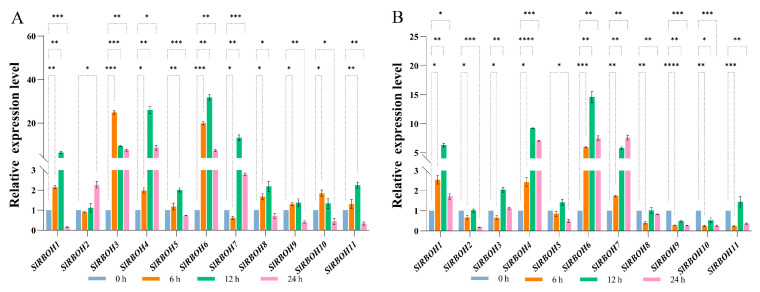
Expression levels of *SlRBOH* genes under (**A**) ABA and (**B**) MeJA. The asterisk (*) indicates that the expression level of the stress group is significantly different from that of the control group (* *p* < 0.05, ** *p* < 0.01, *** *p* < 0.001, and **** *p* < 0.0001, one-way ANOVA, and Tukey test). The samples in the 0 h treatment were used as controls.

**Figure 11 ijms-25-05780-f011:**
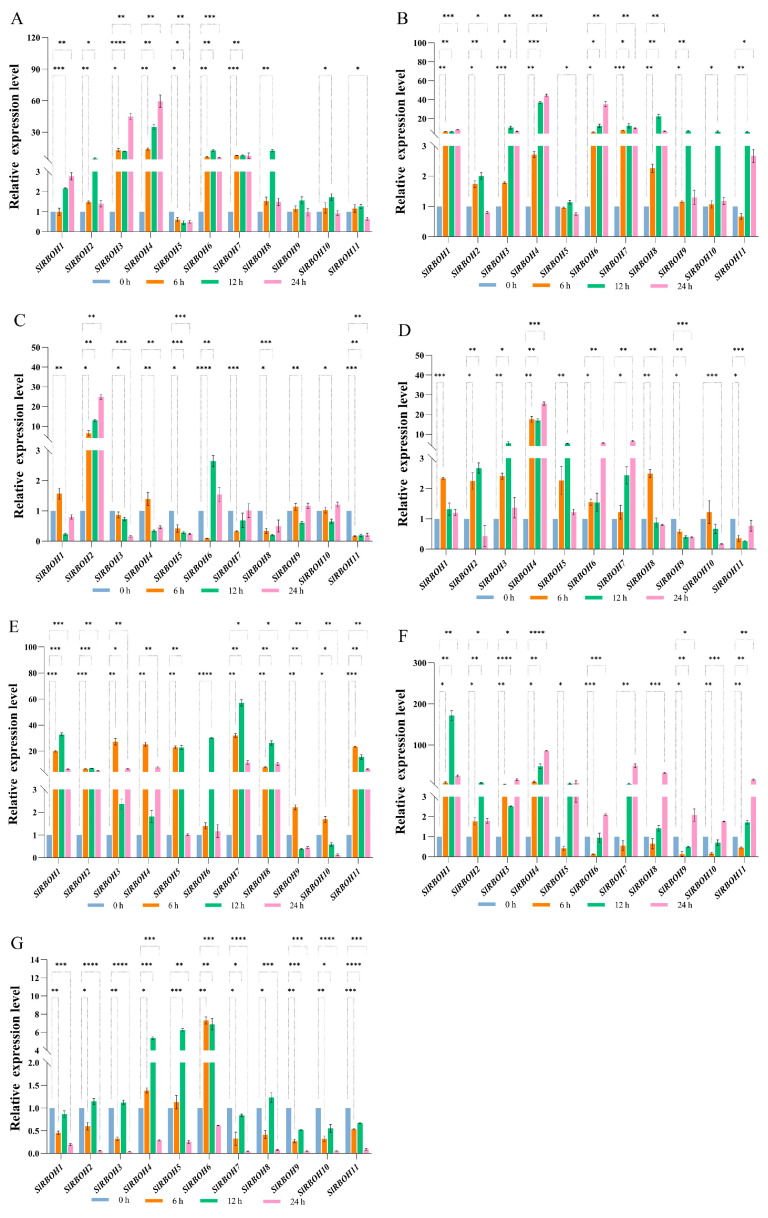
(**A**) NaCl, (**B**) PEG, (**C**) cold, (**D**) heat, (**E**) H_2_O_2_, (**F**) UV, and (**G**) dark period extension treatments. The asterisk (*) indicates the corresponding gene that was significantly up- or down-regulated compared with the 0 h status (* *p* < 0.05, ** *p* < 0.01, *** *p* < 0.001, and **** *p* < 0.0001, one-way ANOVA, and Tukey test). The samples in the 0 h treatment were used as controls.

**Table 1 ijms-25-05780-t001:** Physical and chemical properties of the tomato *SlRBOH* gene family.

Gene	Gene ID	Gene Locus	ORF (bp)	Amino Acid	Instability Index	Molecular Weight/kDa	pI	Aliphatic Index	Signal Peptide	Transmembrane Domain	Grand Average of Hydropathicity (GRAVY)
*SlRBOH1*	XM_004230184.4	Chr01	2598	865	39.6	98,730.42	8.92	84.28	No	Yes	−0.266
*SlRBOH2*	NM_001247342.2	Chr03	2817	938	37.49	105,307.48	9.13	81.88	No	Yes	−0.311
*SlRBOH3*	XM_004239534.4	Chr05	2643	880	49.74	100,465.8	8.71	90.55	No	Yes	−0.154
*SlRBOH4*	XM_019213765.2	Chr05	2571	856	50.71	97,500.21	8.58	91.04	No	Yes	−0.155
*SlRBOH5*	XM_004241593.4	Chr06	2574	857	45.03	97,729.55	9.09	85.01	No	Yes	−0.294
*SlRBOH6*	XM_019214404.2	Chr06	2493	830	41	95,392.74	9.11	80.37	No	Yes	−0.267
*SlRBOH7*	XM_026032004.1	Chr07	2856	951	46.71	107,414.41	8.91	88.63	No	Yes	−0.182
*SlRBOH8*	XM_019214972.2	Chr07	2841	946	46.82	106,813.63	8.75	89.1	No	Yes	−0.172
*SlRBOH9*	NM_001374505.1	Chr08	2892	963	49.59	109,078.14	9.14	88.13	No	Yes	−0.215
*SlRBOH10*	NM_001247197.3	Chr08	2970	989	50.43	112,288.95	9.28	89.26	No	Yes	−0.212
*SlRBOH11*	XM_004251404.4	Chr11	2331	776	41.32	89,066.78	8.82	83.44	No	Yes	−0.196

Gene IDs were derived from NCBI’s gene IDs. The physicochemical properties of the genes were calculated using TBtools. Yes: the sequence has a signal peptide or transmembrane domain. No: the sequence does not have a signal peptide or transmembrane domain.

**Table 2 ijms-25-05780-t002:** Subcellular localization prediction of the tomato SlRBOH members.

Gene	Plasma Membrane	Endoplasmic Reticulum	Nucleus	Peroxisome	Chloroplast	Mitochondrion	Cytosol
*SlRBOH1*	12		2				
*SlRBOH2*	14						
*SlRBOH3*	11	1				2	
*SlRBOH4*	10	1		2		2	
*SlRBOH5*	13		1				
*SlRBOH6*	10	1	3				
*SlRBOH7*	8	1		1	2	2	
*SlRBOH8*	8			1	2	3	
*SlRBOH9*	9	2	1	1	1		
*SlRBOH10*	9	2	1	1	1		
*SlRBOH11*	13						1

**Table 3 ijms-25-05780-t003:** Ten conserved motif sequences of the tomato SlRBOH proteins.

Motif	Width (aa)	Motif Sequence
Motif 1	50	AYFYWVTREQGSFDWFKGVMNEVAEMDHKGVIEMHNYLTSVYEEGDARSA
Motif 2	50	KPPGFKYKSGQYIFVQCPAISSFEWHPFSITSAPGDDYLSVHIRTLGDWT
Motif 3	50	LTGFNAFWYSHHLLIIVYILLIVHGTFLYLVHEWYKKTTWMYJAVPLJLY
Motif 4	50	LFTWKFLQYKQKAAFQVMGYCLATAKGAAETLKFNMALILLPVCRNTJTW
Motif 5	50	VBQRGLPKLLIDGPYGAPAQDYKKYDVLLLVGLGIGATPFISILKDLLNN
Motif 6	50	MVQALNHAKNGVDIVSGTRVRTHFARPNWREVFKRIALKHPYSRIGVFYC
Motif 7	50	HEFWEQISDQSFDARLQIFFDMVDKNGDGKITEEEVKEJIMLSASANKLS
Motif 8	50	TRLGLJIPFDDNINFHKIIAYAIAVGIJJHALNHLACDFPRLINASPEKF
Motif 9	50	DFDEQKPSYIDLVKGVEGVTGIVMVILMAIAFTLATRWFRRNVIKLPKPF
Motif 10	41	KLKEQAAEYAALIMEELDPENLGYIEJWQLETLLLQRDNYM

Width (aa): number of amino acids included in the motif. The results were obtained by MEME.

**Table 4 ijms-25-05780-t004:** Secondary structure of the tomato SlRBOH proteins.

Protein	Alpha Helix (%)	Extended Strand (%)	Beta Turn (%)	Random Coil (%)	Distribution of Secondary Structure Elements
SlRBOH1	44.74	16.76	5.32	33.18	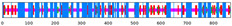
SlRBOH2	45.95	13.11	4.26	36.67	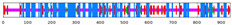
SlRBOH3	45.62	15.64	5.37	33.37	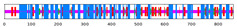
SlRBOH4	44.65	14.64	4.98	35.72	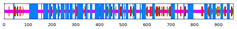
SlRBOH5	44.19	14.16	4.95	36.70	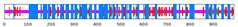
SlRBOH6	45.85	12.93	4.52	36.70	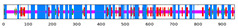
SlRBOH7	43.23	13.85	4.02	38.90	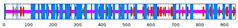
SlRBOH8	44.89	15.34	5.11	34.66	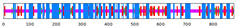
SlRBOH9	44.34	14.10	5.42	36.14	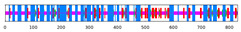
SlRBOH10	50.90	14.05	2.71	32.35	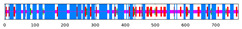
SLRBOH11	43.36	15.07	5.26	36.21	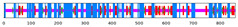

The different secondary structures are expressed as percentages. In the secondary structure diagram, blue represents the alpha helix; green means beta turn; red indicates the extended chain; and pink indicates random coils.

**Table 5 ijms-25-05780-t005:** Functions of the *cis*-acting elements of each gene in the tomato *SlRBOH* genes family.

Cis-Element	Number of Genes	Sequence of Cis-Element	Functions of Cis-Elements
ABRE	16	ACGTG	*cis*-acting element involved in abscisic acid responsiveness
CGTCA-motif	2	CGTCA	*cis*-acting regulatory element involved in MeJA-responsiveness
MRE	6	AACCTAA	MYB binding site involved in light responsiveness
I-box	2	GGATAAGGTG	part of a light-responsive element
AE-box	56	AGAAACTT	part of a module for light responsiveness
TGACG-motif	2	TGACG	*cis*-acting regulatory element involved in MeJA-responsiveness
G-box	2	TACGTG	*cis*-acting regulatory element involved in light responsiveness
ARE	3	AAACCA	*cis*-acting regulatory element essential for anaerobic induction
TCA-element	18	CCATCTTTTT	*cis*-acting element involved in salicylic acid responsiveness
Box 4	12	ATTAAT	part of a conserved DNA module involved in light responsiveness
P-box	9	CCTTTTG	gibberellin-responsive element
MBS	4	CAACTG	MYB binding site involved in drought inducibility
GA-motif	4	ATAGATAA	part of a light-responsive element
GT1-motif	4	GGTTAAT	light-responsive element
TCT-motif	2	TCTTAC	part of a light-responsive element
AT1-motif	3	AATTATTTTTTATT	part of a light-responsive module
LTR	16	CCGAAA	*cis*-acting element involved in low-temperature responsiveness
GATA-motif	2	AAGATAAGATT	part of a light-responsive element
TATC-box	6	TATCCCA	*cis*-acting element involved in gibberellin responsiveness
TGA-element	17	AACGAC	auxin-responsive element
TCCC-motif	2	TCTCCCT	part of a light-responsive element
G-Box	6	CACGTT	*cis*-acting regulatory element involved in light responsiveness

Number of genes: total number of such cis-acting elements contained in tomato *SlRBOHs*.

## Data Availability

All data, tables, and figures in this manuscript are original and are contained within the article and the Appendix A.

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
