# Peer review of "The Characteristics and Expression Analysis of the Tomato SlRBOH Gene Family under Exogenous Phytohormone Treatments and Abiotic Stresses"

_ijms, 2024, doi:10.3390/ijms25115780_

Round 1

Reviewer 1 Report

Comments and Suggestions for Authors

The authors make a complete bioinformatic analisys followed by a expression analysys of the whole family of genes. Both studies are very complete, but there is some more information that should have been included in the discussion to get more outcomes from the available data. The main question is which isoforms are the original or primitive ones and which are originated from recent genome duplication events or gene duplication. This information should be clearly stated in the discussion. Figure 5 would be more useful as part of figure 1, to understand the origins of the current localization of the genes.

Another major problem is with the qRT-PCR data. Camparing one panel to the other is difficult, as each pannel is produced by a different set of primers, with different efficiency. But relative units should be referred to one and in those figures the scale is different in each panel. Given that there is relative expression units, refer the control to 1 in each panel in figures 9, 10 and 11.

Line 19: authors say that genes are "randomly distributed" in the tomato genome. The term randomly is incorrect in this context, as the observed distribution is the result of an evolution process, and this term is in contradiction with figure 5.

Line 96: the expression "sequence logos" is not quite common. Use a standard one such as "domains".

Line 327: play a key role in signal transduction in cells.  

Line 19: authors say that genes are randomly distributed in the tomato genome. The term randomly is incorrect in this context, as the bserved distribution ids the result of an evolution process, and this term is in contradiction with figure 5

Comments on the Quality of English Language

English needs a general revision, there are several grammar mistakes.

Author Response

Dear Reviewer,

Thank you for your valuable feedback on our manuscript entitled "Characteristics and Expression Analysis of the Tomato SlRBOH Family Genes under Exogenous Phytohormones and Abiotic Stresses". We appreciate your time and effort in reviewing our work.

We sincerely appreciate your guidance in improving our manuscript. We have revised the manuscript, and would like to submit it for your consideration. According to your comments and suggestions, we have made corresponding changes.  In addition, the whole manuscript has been read by Dr. Mohammed Mujitaba Dawuda who is from English speaking country (Department of Horticulture, FoA, University for Development Studies, P. O. Box TL 1882, Tamale, Ghana). He gave us many helpful comments and corrections. Please find the detailed responses in the attachment file and the corresponding revisions/corrections highlighted in the re-submitted files.

We would like to express our sincere thanks again to you for the constructive and positive comments.

With best wishes,

Yours sincerely,
Yuanhui Wang, Chunlei Wang

Reviewer 2 Report

Comments and Suggestions for Authors

The current paper devoted to investigation of tomato SLRBOH genes under different conditions.

These genes play an important role in development, therefore investigation of expression patterns in tomato is quite hot topic.

Despite of general interest, paper require significant corrections.

Line 3 (title) is not appropriate. What did authors mean as “under phytohormones”?? Phytohormones is a plant “metabolite” and plant lacking phytohormone not exist. Maybe authors mean exogenous treatments?

Please, clarify.

Line 11: NADPH oxidases are not main ROS producing hormones in plants. The main ROS production source is photosynthesis and electron transport in chloroplasts.

Line 13: “phylogenetic developmental analysis” ??

Line 18: Under different phytohormones?? which one? Endogenous?

Lines 26-27: completely mess.

Line 29: reactive oxygen species – redundant.

Lines 70 – 72: please, split.

Line 104-114: I am not sure average have sense in these contents.

Line 115: structure = domain.

Lines 271 – 282: Do you mean exogenous hormone application? please, provide dose-response curve and IC50 of ABA and MeJ effect. 100 mM (0,1 M) is pretty high, indeed. Curve is required.

Similar for other treatments. IC50 are required!

Moreover, figure 10 demonstrated tissue specific (cell type specific will be better). This is not combined with figure 11 – 12 when only aboveground tissue were used and higher expression in the root were found. As I understand from M&M, primary target of the treatment is root, while RNA is form shoot?

Fig.12 G: what is dark treatment?¿ Plants already have light/dark cycle. Please, clarify what do ypunas 6-12 hours dark? At which moment treatments start? In the end of dark period? It is important point!

Moreover, the cell type in the frame of the organ is a key. Some RBOH may expressed in trichoblast cell only, while another in cortex II-cortex III. These expressions have different effect of stress-resistance, plant growth etc.

Line 342: “syudy”¿??

Line 537: please, provide light in micromole.

Which cycle? 16/8h¿??

Line 539: which hydroponics solution?

Line 543: Under = for induction.

Line 544: 2,94 M H2O2 is very high. IC50 curve are required for each treatment.

Comments on the Quality of English Language

moderate corrections

Author Response

(The authors gave the same response as above.)

Round 2

Reviewer 1 Report

Comments and Suggestions for Authors

Authos have improived the manuscript accoriding to my comments. Now the paper is suitable for acceptance.

Author Response

General comment:

Authos have improived the manuscript accoriding to my comments. Now the paper is suitable for acceptance.

Response:

Thank you very much for your recent feedback on our manuscript. We are delighted to hear that the revisions made according to your comments have brought the paper to a level suitable for acceptance. We have thoroughly addressed each of your suggestions and believe that these modifications have significantly improved the manuscript. Your expert insights were invaluable to refining our work and ensuring its quality and coherence. We are grateful for the time and effort you dedicated to reviewing our manuscript and for your constructive and positive comments. It has been an enriching process for us, and we appreciate your support in enhancing the presentation and content of our paper. We hope that the manuscript now meets all the criteria for publication, and we look forward to the possibility of our research contributing to the journal. Thank you once again for your guidance and support.

Reviewer 2 Report

Comments and Suggestions for Authors

Thank you very much for your constructive response. The text is much better. Maybe in the title you can add under exogenous phytohormone treatments.

The rest part is OK, except minor edition M&M like "dark period extension", not darktreatment.

And some similarvpolishing.

My best regards!

Comments on the Quality of English Language

Proof reading

Author Response

Comment 1:

Thank you very much for your constructive response. The text is much better. Maybe in the title you can add under exogenous phytohormone treatments.

Response:

Thank you for your constructive feedback and your suggestion regarding the title of our manuscript. We have revised the title as you recommended. We appreciate your insightful comments, which have significantly contributed to the improvement of our manuscript. Thank you once again for your guidance and support throughout the review process. The specific modifications are as follows:

Lines 2-4: “Characteristics and Expression Analysis of the Tomato SlRBOH Family Genes under Exogenous Phytohormones and Abiotic Stresses” was changed to “Characteristics and Expression Analysis of the Tomato SlRBOH Family Genes under Exogenous Phytohormones Treatments and Abiotic Stresses”.

Comment 2:

The rest part is OK, except minor edition M&M like "dark period extension", not darktreatment. And some similarvpolishing.

Response:

Thank you for your constructive comments and for pointing out the specific terminology adjustments needed in our manuscript. We have carefully revised these terms and polished the manuscript to address these and similar points you highlighted. We appreciate your attention to detail and your assistance in enhancing the precision of our language and presentation. Your feedback has been instrumental in improving the overall quality of our paper. Thank you once again for your valuable insights. The specific modifications are as follows:

Line 30: “darkness” was changed to “dark period extension”.

Line 287: “Expression levels of SlRBOH genes under ABA (A) and MeJA (B).” was changed to “Expression levels of SlRBOH genes under (A) ABA and (B) MeJA.”.

Line 293: “Cold” was changed to “cold”.

Lines 293,309: “Heat” was changed to “heat”.

Lines 293,320,322,326: “Dark” was changed to “dark period extension”.

Lines 326-327: “NaCl (A), PEG (B), cold (C), heat (D), H2O2 (E), UV (F), and Dark (G) treatments.” was changed to “(A) NaCl, (B) PEG, (C) cold, (D) heat, (E) H2O2, (F) UV, and (G) dark period extension treatments.”.

Line 570: “dark” was changed to “dark period extension”.

Comment 3:

Comments on the Quality of English Language: Proof reading

Response:

Thank you for your comments on the quality of the English in our manuscript. We have taken your feedback seriously and have carefully proofread and revised the text to ensure that the presentation is clear and correct. We appreciate your attention to detail and your commitment to improving the quality of our work. We believe that these revisions have greatly improved the readability and professionalism of the manuscript. Thank you again for your valuable comments and guidance. The specific revisions are listed below:

Line 27: “SlRBOH was” was changed to “SlRBOHs were”.

Line 54: “RBOHS” was changed to “RBOHs”.

Line 56: “Arabidopsis” was changed to “Arabidopsis thaliana”.

Line 60: “Arabidopsis thaliana” was changed to “A. thaliana”.

Line 108: “gene” was changed to “genes”.

Lines 185,192,199: “protein” was changed to “proteins”.

Line 368: “lack” was changed to “lacked”.

Lines 470,473: “RBOH” was changed to “SlRBOH genes”.

Line 476: “RBOH” was changed to “SlRBOH members”.

Lines 577: “and the corresponding IC50 curves can be found in file S1.Also,” was changed to “and the corresponding IC50 curves can be found in file S1. Also,”.